# Characteristics of 9-1-1 Calls Associated with an Increased Risk of Violence Against Paramedics in a Single Canadian Site [note 1]

**DOI:** 10.3390/healthcare13151806

**Published:** 2025-07-25

**Authors:** Justin Mausz, Mandy Johnston, Alan M. Batt, Elizabeth A. Donnelly

**Affiliations:** 1Peel Regional Paramedic Services, 1600 Bovaird Drive East, Brampton, ON L6V 4R5, Canada; mandy.johnston@peelregion.ca; 2Temerty Faculty of Medicine, Department of Family and Community Medicine, University of Toronto, 500 University Avenue, Toronto, ON M5G 1V7, Canada; 3School of Nursing, Queen’s University, 99 University Avenue, Kingston, ON K7L 3N6, Canada; alan.batt@queensu.ca; 4Institute of Health Policy, Management and Evaluation, University of Toronto, 155 College Street, Suite 425, Toronto, ON M5T 3M6, Canada; 5Department of Paramedicine, Building H, Monash University, Peninsula Campus, 47-49 Moorooduc Hwy, Frankston, VIC 3199, Australia; 6School of Social Work, Room 213, University of Windsor, 167 Ferry Street, Windsor, ON N9A 0C5, Canada; donnelly@uwindsor.ca

**Keywords:** paramedics, emergency medical services, violence, workplace violence, occupational health and safety

## Abstract

**Background/Objectives**: Violence is a significant occupational health issue for paramedics, yet underreporting limits efforts to identify and mitigate risk. Leveraging a novel, point-of-event violence reporting system, we aimed to identify characteristics of 9-1-1 calls associated with an increased risk of violence in a single paramedic service in Ontario, Canada. **Methods**: We retrospectively analyzed all electronic violence and patient care reports filed by paramedics in Peel Region and used logistic regression to identify call-level predictors of any violence and, more specifically, physical or sexual assault. **Results**: In total, 374 paramedics filed 974 violence reports, 40% of which documented an assault, corresponding to a rate of 4.18 violent encounters per 1000 9-1-1 calls. In adjusted models, the risk of violence was elevated for calls originating from non-residential locations (e.g., streets, hotels, bars), occurring during afternoon or overnight shifts, and involving young or working-age males. Presenting problems related to intoxication, mental health, or altered mental status were strongly associated with increased risk, with particularly high adjusted odds ratios for assault. **Conclusions**: These findings support the utility of near-miss and violence surveillance systems and highlight the need for multidisciplinary crisis response to high-risk calls, especially those involving mental health or substance use.

## 1. Introduction

### 1.1. Problem Context: Violence Against Paramedics

Occupational violence in paramedicine is increasingly recognized as a growing and complex—yet underreported [1,2,3]—problem, creating the potential for significant harm [4,5,6]. There is abundant research indicating that paramedics are regularly exposed to violence in the course of their work [1,2,3,4,7,8,9,10,11]. As a result, paramedics experience high rates of physical injuries from violence—often surpassing injury rates for other public safety personnel, like firefighters [12]. Exposure to violence also has significant impacts on mental health, job satisfaction and well-being among paramedics [5,7,11,13,14,15].

### 1.2. Paramedic Mental Health

Given high rates of posttraumatic stress disorder [16], burnout [17,18], depression and anxiety [16,19] among paramedics, exposure to violence has the potential to compound an existing mental health crisis in the profession [20] and worsen the workforce shortages being experienced within healthcare more broadly [21,22].

There is, then, a pressing need to develop strategies that both identify and mitigate the risk of violence during 9-1-1 calls, but any meaningful effort at risk mitigation must be informed by accurate event-level data about violent encounters.

### 1.3. Pervasive Underreporting

Unfortunately, there is widespread recognition that violent encounters are chronically underreported, with only the most serious (i.e., those resulting in significant physical injuries) documented through formal incident reporting processes [1,2,3]. The reasons for this are multifaceted, but commonly include administrative barriers, such as overly onerous incident reporting processes [7], underpinned by an organizational culture that considers violence “part of the job” [23] and not worth reporting. In a 2014 study of paramedics from two Canadian provinces, Bigham and colleagues found that paramedics rarely reported incidents of violence to police or service leadership teams and lacked culturally acceptable reporting pathways [7]. Among other things, Bigham et al. suggested that reporting might be improved if paramedics were involved in the development of purpose-built incident reporting processes [7]. To address this issue, we developed a new point-of-event reporting system, which forms the basis of the present study.

### 1.4. The External Violence Incident Report

As part of a broader workplace violence prevention program within the paramedic service, our team developed a new, streamlined violence reporting process. The External Violence Incident Report (EVIR) is a purpose-built incident report developed in consultation with paramedics to gather quantitative and qualitative data about violence, including the type (e.g., verbal abuse, physical assault) and source (e.g., patient, bystander) of violence during 9-1-1 calls. Additional details are provided below.

#### 1.4.1. The Benefits of Event-Level Data

In contrast to survey methods [4,7,24,25,26] or injury database searches [12,27,28] that have been the mainstay of research on violence against paramedics, the collection of event-level data about violent encounters as part of routine clinical and administrative documentation allows for more granular epidemiological study of the problem. This is particularly important in developing upstream risk stratification strategies.

#### 1.4.2. Study Objectives

Whereas previous studies have relied on retrospective or otherwise limited datasets, this study introduces an approach based on real-time, event-level reporting that allows for a more detailed analysis of risk factors. Therefore, leveraging this novel incident reporting process, our objective was to identify types of 9-1-1 calls that are associated with an increased risk of violence against paramedics.

## 2. Materials and Methods

### 2.1. Overview

We conducted this study as part of a larger research program on violence against paramedics, and we detailed our overall approach in an earlier publication [29]. For this study, we retrospectively reviewed electronic Patient Care Records (ePCRs) and External Violence Incident Reports (EVIRs) filed over a two-year period since the launch of the new reporting process in a single paramedic service in Ontario, Canada. Using logistic regression analyses, our objectives were to identify covariates in ePCR data that are associated with an increased risk of any violence and (as a subgroup) of a physical or sexual assault on a paramedic.

### 2.2. Setting and Target Population

This research is situated in the Regional Municipality of Peel in Ontario, Canada. Peel Regional Paramedic Services (PRPS) is the sole provider of land ambulance and paramedic services to the municipalities of Brampton, Mississauga, and Caledon, collectively encompassing the Region of Peel. The paramedic service is publicly funded, provincially regulated, and separate from police and fire services. PRPS serves a population of approximately 1.3 million residents across a mixed urban/suburban geography of 1200 km^2^ with a complement of approximately 750 Primary and Advanced Care Paramedics who respond to an average of 130,000 emergency calls per year—collectively making the service the second largest in the Province of Ontario by staffing and call volume.

The introduction of the new reporting process occurred alongside a suite of initiatives as part of a larger workplace violence prevention program. This included (among other things), new policies and procedures, patient restraint equipment, and a public position statement of ‘zero tolerance’ for violence against paramedics.

### 2.3. Instruments: The EVIR

We described the development of the EVIR in an earlier publication [30]. Briefly, the development phase involved an extensive stakeholder consultation, gap analysis, and pilot testing process. The result was an incident report built to gather comprehensive quantitative and qualitative information about violent encounters with the public during 9-1-1 calls. The report primarily makes use of drop-down menu selection questions to gather information about the type (e.g., verbal abuse, threats, sexual harassment, or physical or sexual assault), circumstances (e.g., alcohol or drug intoxication, mental health crisis, or cognitive impairment), and source (e.g., patient, family member, bystander, etc.) of violence, with one free-text box in which paramedics can type a detailed narrative description of the violent encounter. The form is embedded within the ePCR software (version 2.1.8515, ESO). Provincial documentation standards require paramedics to complete an ePCR after each 9-1-1 call and additionally mandate the completion of incident reports in unusual circumstances, such as situations that involve a risk to the patient or paramedic. The EVIR constitutes an incident report under these standards and the software reminds paramedics to complete an EVIR when filing an ePCR if they experienced any form of violence during the call. The EVIR in its entirety is available as Appendix A.

### 2.4. Additional Data Sources

The ePCRs, meanwhile, gather administrative and clinical data about the call and the treatment provided to patients. For this study, we were interested in the following fields from all ePCRs filed during the study period: dispatch priority (e.g., urgent, non-urgent), location type code (e.g., residence, street/intersection, etc.), primary presenting problem code (e.g., altered level of consciousness, shortness of breath, etc.), patient acuity via the Canadian Triage Acuity Scale (CTAS) [31], patient age and sex, and all call event times (e.g., call received, crew notified, etc.). When an EVIR is filed, these data points are automatically pulled from the corresponding ePCR into the body text of the violence report. The ePCR software incorporates compliance rules for documentation that minimize missing data by making most fields mandatory, depending on the 9-1-1 call type; in practical terms this afforded a very low (i.e., near zero) rate of missing information.

### 2.5. Data Collection Period

Our data collection window spanned a two-year period from 1 February 2021 (when the reporting process was launched) through 31 January 2023. We abstracted the administrative data points described above from all ePCRs filed during the study window and all fields from the EVIR where one was filed for a particular service call. Although not primarily intended for research, the EVIR includes a passive consent process where the documenting paramedic can opt out of having a particular report used for research purposes by ticking a box; in these cases, the reports were excluded.

### 2.6. Inclusion and Exclusion Criteria

All cases in which an ePCR was filed for a 9-1-1 call where the paramedics arrived at the scene were eligible for inclusion in the study. Similarly, all complete EVIRs were eligible for inclusion, except where the documenting paramedic did not consent to use of the report for research purposes. Incomplete records or ePCRs for calls where the paramedic crew did not arrive scene were excluded.

### 2.7. Variables and Outcomes

Our list of covariates included the day of week, time of day, dispatch priority code, call location code, primary presenting problem code, patient acuity level, and patient gender and age. Paramedics in our service normally work 12 h day or night shifts, with additional staff scheduled during ‘peak’ demand periods. However, to allow for a more granular time-of-day analysis, we defined three non-overlapping 8 h buckets: day (06:00–13:59 h), afternoon (14:00–21:59 h), and overnight (22:00–05:59 h) shifts, respectively. Several of the covariates are multi-level categorical variables, including the call location code (26 levels, e.g., residence, street, restaurant) and primary presenting problem code (55 levels, e.g., cardiac arrest, shortness of breath, unconscious).

Our primary outcome was whether an EVIR was filed for a particular 9-1-1 call (‘any violence’), with physical or sexual assault on a paramedic (‘any assault’) as our secondary outcome.

### 2.8. Statistical Procedures

We used descriptive and summary statistics to report on the characteristics of the data, including the types of violence reports included in the study. Using complete case analysis, we used logistic regression to estimate the probability of our primary and secondary outcomes using the covariates described above, as these are characteristics of 9-1-1 calls that are generally available to the paramedics at the point of dispatch.

An overview of our analytical approach is provided in Figure 1. Because of the large number of variables (and levels of variables) included in the analysis, our approach to modeling involved several iterative steps to first reduce the number of covariates that would be included in the final models. First, we ran cross-tabulations on each covariate to identify the number of EVIRs associated with the variable and each of its levels (where applicable); covariates with observed frequencies of at least 10 events were retained for inclusion. Next, and where applicable, we grouped similar variable levels together (e.g., ‘apartment/condo’ and ‘house’ was reclassified as ‘private residence’); in this manner, ‘call location code’ was reduced from 26 levels to 7 in a new composite ‘high risk call location’ variable. Primary presenting problem was similarly reduced from 56 levels to 8 in a new ‘high risk problem’ variable. We did this to improve model stability while preserving statistical power. A step-by-step accounting of this reclassification process is offered as Appendix A.

With variable reclassifications complete, we ran univariate analyses to test the association between each new variable and our primary outcome (‘any violence’). Covariates or levels of categorical variables that did not reach significance at *p* < 0.05 were dropped. Significant covariates were then entered as a group into a new model to test the main effects. Again, covariates or levels of categorical variables that did not retain significance at *p* < 0.05 at this stage were dropped. Finally, we constructed interaction terms of significant covariates that had plausibility for presenting an increased risk of violence and introduced these terms individually into our adjusted model. This process was repeated for our secondary outcome of any physical or sexual assault.

The principal assumptions of concern for this analysis were collinearity and overfitting; our approach was intended to minimize both risks by collapsing related variables and filtering covariates through a robust selection process. Only cases with complete data for the variable of interest were used. In constructing our models, our interest was in the significance of the individual covariates rather than the predictive capacity of the model as a whole. Our threshold for significance of the individual covariates was a *p*-value less than 5% and 95% confidence intervals that excluded the null. All analyses were carried out in SPSS Statistics (Version 29).

### 2.9. Research Ethics

This study was approved by the University of Toronto Research Ethics Board (protocol #44162, 3 May 2023).

## 3. Results

### 3.1. Overview

Over the two-year study period, the paramedic service generated a total of 251,628 ePCRs. After excluding duplicate forms filed for the same incident, this left 224,739 unique 9-1-1 calls for analysis and 985 EVIRs, of which, 44 (4.4%) refused consent for use in research and were excluded. The final sample included 941 EVIRs filed by 374 paramedics for an overall rate of violence of 4.18 incidents per 1000 9-1-1 calls. Of the reports, 39% (n = 364) documented some form of physical or sexual assault, either alone or in combination with other forms of violence, yielding a rate of 1.61 assaults per 1000 calls. In total, 81 incidents resulted in a paramedic being physically harmed during the encounter.

The majority (n = 697; 74%) of the violence occurred at the emergency scene and patients were cited as the perpetrator in 83% of cases (n = 781). Alcohol or drugs (n = 461; 49%) or mental health concerns (n = 317; 37%) were frequently listed as contributing circumstances. The reports were fairly uniformly distributed across days of the week, with the lowest proportion (12%) on Monday and the highest (16%) on Tuesday and no statistically significant differences between days (*p* = 0.618). The highest proportion of reports (47%) was filed during the afternoon shift.

### 3.2. Univariate Analyses

In our univariate analyses, dispatch priority (urgent), location code (residence, store, long-term care home, hotel, street, or restaurant), shift (afternoon and overnight), and several primary presenting problems (e.g., intoxication, mental health, trauma, chest pain, and altered mental status) were all individually associated with an increased risk of any violence. Compared to ‘non-urgent’ acuity (CTAS 5), patients requiring resuscitation (CTAS 1; Odds Ratio [OR] 1.64, 95% Confidence Interval [CI] 1.02–2.66) or emergent care (CTAS 2; OR 2.02, 95% CI 1.41–2.88) were associated with an increased risk of violence, while ‘urgent’ (CTAS 3) or ‘less urgent’ (CTAS 4) acuity levels were not; this variable was reclassified into a ‘High Acuity’ (CTAS 1 or 2) variable.

Parameters for age brackets were constructed using Statistics Canada Life Cycle Groupings, with seniors (aged 65+) as the reference category. All levels were significantly associated with the risk of violence, with elevated risks among youth and working aged patients. These variables were entered into an omnibus main effects multivariable model (analysis step 3) for filtering (Table 1).

While patient sex was not significantly associated with the risk of violence on its own, when stratified by life cycle grouping, an effect emerged suggesting that 9-1-1 calls involving young and working aged men were associated with an increased risk of violence (Table 2). These parameters were reclassified into a 3-level composite patient demographic variable for inclusion in adjusted models.

### 3.3. Adjusted Models

After entering the variables as described above into an omnibus main effects model, dispatch priority (urgent: aOR 0.92, 95% CI 0.80–1.06), as well as some levels of the location variable (store: aOR 1.32, 95% CI 0.92–1.88; and long-term care home: aOR 1.33, 95% CI 0.90–1.96) and primary problem variable (chest pain: aOR 1.21, 95% CI 0.85–1.72) were no longer significant at the *p* < 0.05 threshold. These covariates were dropped.

Introducing an interaction term for call location by primary presenting problem yielded no covariates significant at *p* < 0.05 and resulted in some levels of the location code losing significance in the more complicated model. The same was true for introducing a term for shift by call location; in both cases, the terms were dropped. However, introducing a term for high acuity by primary presenting problem indicated that patients with a CTAS score of 1 or 2 with a primary presenting problem related to mental health (‘behavioral/psychiatric problem’ per the ePCR code) were associated with an increased risk of violence (aOR 3.54, 95% CI 2.47–5.08). Importantly, the directionality of this effect is questionable, given that triage guidelines score violent patients with mental health concerns higher. The term was not significant for other variable levels.

The final adjusted model is presented in Table 3. Calls occurring in restaurants or bars (aOR 2.84), hotels (aOR 1.50), or on a public street (aOR 1.33) were significantly associated with an increased risk of violence compared to other locations. Similarly, calls occurring during afternoon (aOR 1.26) or overnight shifts (aOR 1.25) were associated with an increased risk of violence compared to day shifts. Calls involving working age men as patients also presented an increased risk of violence compared to other age and gender strata (aOR 1.78). However, the primary presenting problem was the most significant predictor of the risk of violence, with calls involving alcohol or drug intoxication (aOR 9.06) or mental health-related concerns (aOR 10.59) presenting an especially increased risk of all forms of violence after adjusting for other call characteristics.

### 3.4. Secondary Outcome: Any Assault

We repeated this process for our secondary outcome of any physical or sexual assault, beginning first with the covariates identified during our univariate analyses as associated with an increased risk of any violence (Table 1 and Table 2) entered into an omnibus main effects model.

The details of variables included in the final model are presented in Table 4. While many of the same predictors remained significant, the magnitude of the risk was notably higher for several covariates compared to the broader model for all forms of violence. Calls involving mental health concerns (aOR 30.37) and alcohol or drug intoxication (aOR 21.37) had particularly elevated odds of assault—approximately triple the corresponding estimates in the model for any violence. Similarly, altered mental status (aOR 6.06) and trauma-related problems (aOR 2.19) were also associated with an increased risk of assault.

In contrast to the model for any violence, long-term care homes emerged as a strong contextual predictor for assault (aOR 2.51), whereas the public locations identified as high risk in the broader model were not retained. High acuity remained significantly associated with an increased risk of assault (aOR 2.86), but shift type was more modest, with an increased risk attributed to afternoon (aOR 1.28) but not overnight shifts. Because of the smaller number of events, we did not test for interaction effects between covariates in our secondary outcome analysis.

## 4. Discussion

Our objective in this study was to identify characteristics of 9-1-1 calls that are associated with an increased risk of violence and, more specifically, with an increased risk of physical or sexual assault on a paramedic. We found that calls occurring in public locations (e.g., hotel, street, restaurant/bar), during afternoon or overnight shifts, or involving young or working age males presented an increased risk of all forms of violence. The risk, however, appeared to be largely driven by the primary presenting problem, particularly for problem codes related to substance use—such as alcohol or drug (including opioid) intoxication—and mental health-related concerns. In our adjusted models, these problem types were associated with a 9- and 10-fold increase in the odds of all forms of violence, respectively. The risk was even more pronounced for cases involving physical or sexual assault on a paramedic. These findings have important implications for research and policy.

First, and from a research perspective, the rates of violence we observed are lower than have been found in a recent study involving 15 Emergency Medical Services (EMS) agencies in the Midwest United States [9]. Using a checkbox reporting process embedded in the ePCR, McGuire and colleagues recorded 8.6 violent encounters per 1000 9-1-1 calls and 3.7 assaults per 1000 9-1-1 calls [9]—rates approximately double those in our study. One plausible explanation for the difference is the administrative burden associated with the respective reporting processes. In our study, the paramedics completed a full incident report with several drop-down menu selection questions, in addition to free text fields. The ePCR software prompts the paramedics to complete the report if they experienced violence during the call but the report itself is not mandatory. In McGuire’s study, however, the reporting process involved a single mandatory question on the ePCR [9]. Given pervasive underreporting [7,9,23], reducing the administrative burden to document violent encounters appears to be important. Merging the two approaches such that every ePCR has a mandatory filtering question about violence could be useful, with affirmative responses then generating a more detailed—but streamlined—incident report. For international readers looking to adopt the EVIR, a detailed description is included in an earlier publication [30]. Briefly, we recommend a gap analysis of existing reporting processes to ensure alignment within the regulatory environment followed by stakeholder consultations and pilot testing. For the reporting process to gain widespread acceptance among the workforce, the reports need to be seen to lead to meaningful improvements in provider safety [32].

Second, like our study, McGuire’s recent work also pointed to 9-1-1 calls involving male patients, alcohol or drug use, and emotional distress, behavioral, or psychological problems as presenting an increased risk of violence [9]. Similarly, a recent study by Kim and colleagues of two US EMS agencies demonstrated that violent 9-1-1 calls were most commonly associated with psychiatric-related presenting problems [33]. These trends echo media reporting in the UK pointing to accelerating rates of violence against ambulance staff, with substance use or mental health-related concerns cited as contributing factors [34,35]. In our data, calls where the primary presenting problem was related to mental health or substance use concerns accounted for approximately 7% of the overall call volume but were listed as the primary problem in 60% of documented assaults. Meanwhile, sudden cardiac arrest—around which most modern paramedic services are optimized—accounted for less than 1% of calls during the study period. This underscores a well-documented imbalance between paramedic service delivery and community need [36,37] but suggests the misalignment problem may be contributing to an elevated risk of occupational violence.

Recent qualitative research has begun to shed light on the potential safety implications—for both providers and patients—of this misalignment problem. For example, Drew and colleagues identified that resource, legislative, and training constraints mean Australian paramedics are often ill-prepared to respond to calls involving mental health problems, creating tension at the scene that can escalate into violence [38]. Similarly, work by Ford-Jones and Bolster underscored that entry-to-practice training, practice guidelines, and policy documents often prime paramedics to expect violent behavior from patients accessing 9-1-1 for mental health or substance use concerns [39,40,41]. Combined with a lack of suitable treatment and disposition pathways and specialist clinicians, the result, again, is that paramedics lack the ability to safely engage with patients with mental health concerns, creating an increased risk for violence against the paramedics and sub-optimal care for patients [41]. Finally, a recent scoping review by Bolster and colleagues on the evolving paramedic role in the opioid crisis underscored the need for paramedic services to provide culturally competent, holistic patient care for people who use drugs [39].

Taken together, the extant research suggests that (1) 9-1-1 calls involving mental health and substance use are highly prevalent [39,41,42]; (2) gaps in paramedic training, clinical practice guidelines, and regulatory frameworks mean paramedics are both individually and organizationally ill-prepared to respond to these call types [38,40,41]; and (3) the resulting misalignment in service delivery poses legitimate safety risks to paramedics [9,33], in addition to contributing to poor outcomes for patients [41]. One promising solution is the development of specialized interdisciplinary teams tasked specifically with responding to mental health and substance use crises in the community. Although within the context of 9-1-1 response these teams are commonly deployed using police infrastructure, there is early work on new frameworks for such teams to develop within Ontario paramedic services [43]. Similarly, a recent study examined the impact of a new alternative treatment and disposition pathways for paramedics for patients with mental health concerns [42]. In the initial evaluation, as many as 1 in 3 patients with mental health concerns were eligible to be transported to a specialized mental health crisis center, resulting in a 50% reduction in ambulance offload delay times, with 96% of paramedics rating the program favorably [42]. From a training and policy perspective, our findings suggest that paramedic services should develop protocols to triage high-risk calls for response by specially trained and interdisciplinary mental health crisis teams.

While these programs can reasonably be expected to improve service delivery for patients accessing paramedic services for mental health-related concerns and may mitigate at least some of the violence risk associated with these encounters, occupational violence against paramedics is a complex problem, requiring a multimodal response. Recent studies have proposed body armor for paramedics [10], body-worn and in-vehicle cameras [44], address hazard flagging [45], and high-fidelity simulation training [46] as potential solutions, albeit with mixed results. Ultimately, the research in this area is still developing and additional studies are needed. One promising intervention is a new 3-question violence risk assessment tool developed specifically for use by paramedics. In its derivation phase, the Aggressive Behavior Risk Assessment Tool for Emergency Medical Services (ABRAT-EMS) had a positive predictive value of 34% for identifying potentially violent EMS patients and a negative predictive value of 99% for patients below the “high” risk threshold [33]. This suggests the checklist has potential as a short, point-of-contact violence screening tool. Replicating these findings in other services and adapting these or similar tools for upstream risk assessment during 9-1-1 call-taking is an important avenue for further research. Examining whether certain dispatch classifications are associated with an increased risk of violence would allow for easier and earlier risk mitigation before crews are dispatched to potentially violent scenes.

### Limitations

Our findings should be interpreted within the context of certain limitations. First, as with any retrospective study, the validity of our findings hinges on the quality of the documentation, which can be inconsistent across providers. Second, and on a related note, earlier research has highlighted that occupational violence in healthcare is chronically underreported [47]. This not only means our estimates might be conservative, but also introduces the potential for selection bias in the events that the paramedics feel warrant documenting—potentially inflating the relative risks associated with certain call types. The associations we identified should not be interpreted as indicating causal relationships. Third, the covariates used in our modeling are not always available to the paramedic at the point of call assignment. Ideally, the analysis would have leveraged dispatch triaging tools that classify 9-1-1 calls into predetermined categories; however, the call prioritization software changed midway during our study. Finally, we made no effort to validate the veracity of the paramedics’ claims of violence documented in the reports, although we have no reason to doubt them.

## 5. Conclusions

Out of every one thousand 9-1-1 calls, more than four resulted in a documented incident of violence against the responding paramedics. Violence was more likely to occur during afternoon or overnight shifts, at non-residential locations, and for 9-1-1 calls involving young or working-aged men. Calls related to altered mental status—and particularly those involving mental health or substance use concerns—were among the strongest predictors of violence, with markedly elevated adjusted odds of physical or sexual assault.

Given the risk of both physical and psychological harm to paramedics, our findings underscore the value of the EVIR as a proactive surveillance tool for monitoring and responding to violence as an occupational health threat. From an operational perspective, these results support the development of targeted risk mitigation strategies, including stronger detection and call triaging supports during call handling and dispatch.

Taken together, our findings point to an urgent need for paramedic services to establish formal protocols to triage mental health and substance use-related calls for potential co-response by multidisciplinary mental health crisis teams. These teams may help de-escalate high-risk encounters and reduce the likelihood of violence. Integrated approaches like these, alongside broader violence prevention efforts, are essential to protecting the safety and mental health of paramedics while providing high-quality patient care.

## Figures and Tables

**Figure 1 healthcare-13-01806-f001:**
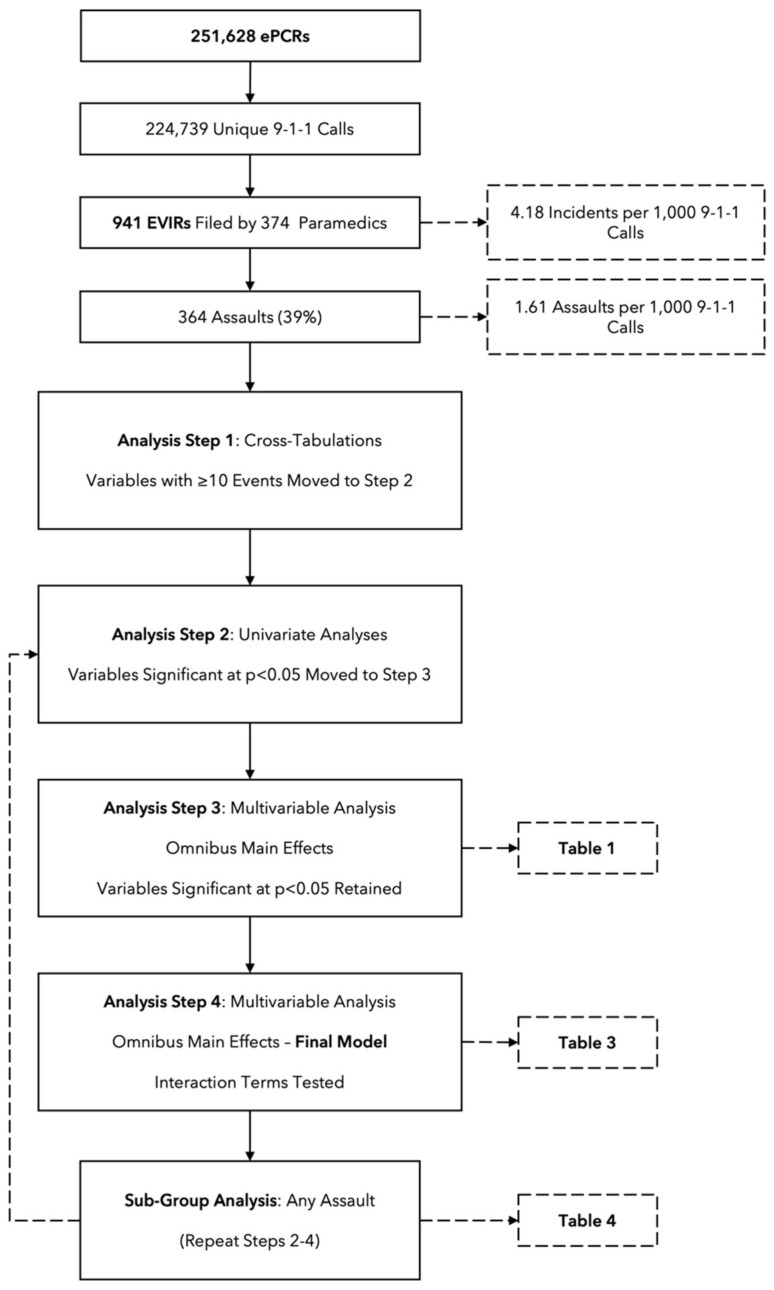
Overview of variable filtering procedures and analysis steps. ePCR = electronic Patient Care Record; EVIR = External Violence Incident Report.

**Table 1 healthcare-13-01806-t001:** Multivariable analyses for any reported violence. Day Shift: 06:00–13:59; Afternoon Shift: 14:00–21:59; Overnight Shift: 22:00–05:59. CTAS = Canadian Triage Acuity Scale. Patient age categories drawn from Statistics Canada life cycle age classification.

Category	Covariate	OR	95% CI	*p*-Value	Decision
Dispatch Priority	Priority 3 (‘Prompt’)	1.21	0.91, 1.62	0.17	Dropped
	Priority 4 (‘Urgent’)	1.34	1.02, 1.77	0.03	Retained
Call Location Code	Other Location (Reference)				
	Private Residence	0.99	0.81, 1.22	0.97	Dropped
	Store	2.02	1.42, 2.87	<0.001	Retained
	Long-Term Care Home	0.82	0.56, 1.20	0.31	Dropped
	Hotel	3.17	2.07, 4.84	<0.001	Retained
	Street/Intersection	1.91	1.50, 2.44	<0.001	Retained
	Restaurant/Bar	5.55	3.58, 8.60	<0.001	Retained
Shift	Day Shift (Reference)				Retained
	Afternoon Shift	1.52	1.30, 1.77	<0.001	Retained
	Overnight Shift	1.59	1.33, 1.90	<0.001	Retained
Acuity	CTAS 5 (Non-Urgent; Reference)				Reclassified into “High Acuity” variable (i.e., CTAS 1 or 2)
	CTAS 4 (Less Urgent)	0.66	0.85, 1.71	0.05	
	CTAS 3 (Urgent)	1.20	0.85, 1.71	0.28	
	CTAS 2 (Emergent)	2.02	1.41, 2.88	<0.001	
	CTAS 1 (Resuscitation)	1.64	1.02, 2.66	0.04	
Primary Problem Code	Other Problem (Reference)				
	General Problem	1.18	0.92, 1.52	0.18	Dropped
	Intoxication	16.54	13.34, 20.48	<0.001	Retained
	Abdominal Pain	1.35	0.95, 1.93	0.09	Dropped
	Mental Health	12.71	10.29, 15.69	<0.001	Retained
	Dyspnea	1.01	0.67, 1.54	0.93	Dropped
	Trauma/Injury	2.02	1.58, 2.58	<0.001	Retained
	Chest Pain	1.58	1.09, 6.56	0.01	Retained
	Altered Mental Status	4.77	3.47, 6.56	<0.001	Retained
Patient Age	Seniors (65+ Years; Reference)				
	Working Age (25–64 Years)	2.40	2.05, 2.80	<0.001	Retained
	Youth (15–24 Years)	1.69	1.31, 2.17	<0.001	Retained
	Children (0–14 Years)	0.38	0.21, 0.68	0.001	Retained

**Table 2 healthcare-13-01806-t002:** Composite variable for patient demographics, cross-referenced with Statistics Canada life cycle age groupings.

Patient Age/Sex Groups	OR	95% CI	*p*-Value
Children—Female	0.42	0.19, 0.90	0.026
Children—Male	0.24	0.10, 0.59	0.002
Youth—Female	1.20	0.83, 1.74	0.324
Youth—Male	1.64	1.18, 2.29	0.003
Working Age—Female	1.22	0.96, 1.54	0.094
Working Age—Male	2.73	2.23, 3.38	<0.001
Senior—Female	0.69	0.53, 0.90	0.007
Senior—Male (Reference)			

**Table 3 healthcare-13-01806-t003:** Final adjusted model for any reported violence.

Variable Category	Covariate	OR	95% CI	*p*-Value
Pick Up Location	Other Location (Reference)			
	Hotel	1.50	1.01, 2.24	0.041
	Street	1.33	1.11, 1,60	<0.001
	Restaurant/Bar	2.84	1.64, 3.76	<0.001
Shift	Day Shift (Reference)			
	Afternoon Shift	1.26	1.08, 1.48	0.003
	Overnight Shift	1.25	1.05, 1.50	0.011
Primary Problem	Other Problem (Reference)			
	Intoxication	9.06	7.49, 10.95	<0.001
	Mental Health	10.59	8.89, 12.61	<0.001
	Trauma	1.67	1.34, 2.08	<0.001
	Altered Mental Status	3.27	2.43, 4.40	<0.001
Acuity	High Acuity	1.86	1.62, 2.15	<0.001
Patient Demographics	Other Age/Gender Group (Reference)			
	Youth—Male	0.31	0.12, 0.75	0.009
	Working Age—Male	1.78	1.54, 2.04	<0.001

**Table 4 healthcare-13-01806-t004:** Final adjusted model for any physical or sexual assault on a paramedic. Day Shift: 06:00–13:59; Afternoon Shift: 14:00–21:59; Overnight Shift: 22:00–05:59.

Variable Category	Covariate	*p*-Value	OR	95% CI
Pick-Up Location	Long-Term Care	<0.001	2.51	1.71, 3.70
Shift	Day Shift (Reference)			
	Afternoon Shift	0.042	1.28	1.00, 1.64
	Overnight Shift	0.693	1.06	0.79, 1.42
Primary Problem	Other Problem (Reference)			
	Intoxication	<0.001	21.37	15.43, 29.58
	Mental Health	<0.001	30.37	22.67, 40.80
	Trauma/Injury	<0.001	2.19	1.47, 3.27
	Altered Mental Status	<0.001	6.06	4.02, 9.14
Acuity	High Acuity	<0.001	2.86	2.30, 3.56
Patient Demographics	Senior—Male (Reference)			
	Children—Female	0.036	0.22	0.05, 0.90
	Children—Male	0.037	0.12	0.01, 0.88
	Youth—Female	0.036	0.57	0.34, 0.96
	Youth—Male	0.318	0.77	0.47, 1.27
	Working Age—Female	0.187	0.78	0.54, 1.12
	Working Age—Male	0.210	0.80	0.56, 1.13
	Senior—Female	0.019	0.63	0.43, 0.92

## Data Availability

Data from this study may be made available to interested researchers on a case-by-case basis, subject to institutional review board approval and a negotiated data sharing agreement with suitable privacy and data security protections.

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
