# Peer review of "Characteristics of 9-1-1 Calls Associated with an Increased Risk of Violence Against Paramedics in a Single Canadian Siteâ€"

_healthcare, 2025, doi:10.3390/healthcare13151806_

Round 1
Reviewer 1 Report
Comments and Suggestions for Authors
This article presents a retrospective observational study using electronic patient care records and a novel External Violence Incident Report system to identify 9-1-1 call characteristics associated with increased risk of violence against paramedics. However, for improvement, the following are suggested:
- Include more discussion on how the findings compare with international EMS contexts.
- Recommend or suggest ways to validate the EVIR tool in other settings.
- I suggest improve the abstract and make it direct to the point.
- Clarify how the results could inform EMS training or policy changes more concretely.
- Provide a detailed methodology so the reader may understand the analysis easily.
- Highlight the exclusion and inclusion criteria in the methodology.
- Explain the odds for the variables in more clear way.
- Separate the future studies from limitations section.
- One of the weaknesses of this paper is that it has no generalizability as is limited to one Canadian region. Explain how we can benefit from the paper results in other countries
- The study does not establish causal relationships for its finding
Author Response
Please refer to the attached file for our detailed response. We are grateful to you for having taken the time to provide feedback on our work.

Reviewer 2 Report
Comments and Suggestions for Authors
Abstract
It would be beneficial to structure the abstract into clear sections: background, methods, results, and conclusions. This would allow for a more organised and reader-friendly summary.
Introduction
The introduction merges several distinct topics (underreporting, organisational culture, psychological impact, etc.) without clear separation.
It is recommended to restructure the introduction into logically organised thematic blocks:
- Problem context (violence against paramedics)
- Documented impacts (mental health, absenteeism, safety)
- Issues with underreporting and data limitations
- The need for new approaches based on event-level data
- Presentation of the study objective
In line 37, the sentence “There is now abundant research indicating that paramedics are regularly exposed to violence…” could be rephrased for clarity and conciseness as:
“Studies show that violence is a frequent reality for paramedics, with significant physical and psychological consequences.”
The introduction mentions underreporting and argues for the importance of higher-quality data, but fails to clearly link this issue to the new reporting system (EVIR).
A transitional sentence could be added such as:
“To address this gap, we developed a new on-scene reporting system – the External Violence Incident Report (EVIR) – which forms the basis of the present study.”
While the introduction references numerous prior studies, it does not critically examine the limitations of previous methods (e.g., surveys, administrative databases) or explain how this study offers a novel contribution.
A more analytical comparison is suggested:
“Whereas previous studies have relied on retrospective or limited datasets, this study introduces an approach based on real-time, event-level reporting, allowing for a more detailed analysis of risk factors.”
Methods
This section includes several overlapping subheadings (“Overview”, “Setting and Context”, “Data Collection”, etc.), making it somewhat dense and difficult to follow.
It is advisable to restructure the section using clearer, non-redundant subheadings, such as:
- Setting and target population
- Data source and collection period
- Instrument (EVIR)
- Variables and outcomes
- Statistical procedures
The article refers to a “passive consent opt-out” but does not state how many reports were excluded on this basis. This should be quantified.
Although paramedics work in 12-hour shifts, the authors chose 8-hour time blocks (morning, afternoon, overnight) for analysis, which may be confusing. The rationale for this choice should be clearly explained.
While logistic regression is mentioned, the statistical software used (e.g., SPSS, R, Stata) is not specified and should be.
In the methods, reference should be made to the approval of the study's ethics committee
Results
The results section relies heavily on tables, with limited narrative synthesis in the text. A clearer summary should be included, highlighting key effects.
For example: “The risk of violence was over ten times higher when the primary complaint was related to mental health (OR = 10.59), indicating this as the highest-risk category in the adjusted model.”
The model for “assault” (physical or sexual) is presented but receives little comparative discussion against the primary outcome. A concise comparison would be helpful: “While general risk factors for violence included shift and location, physical/sexual assault was more strongly associated with mental health issues and intoxication, with odds ratios exceeding 20.”
Conclusion
The conclusion does not sufficiently emphasise the value of the EVIR system as a surveillance tool.
It also lacks clear suggestions for future research or operational recommendations.
The final sentence is vague (“...should consider new approaches…”). It would be stronger as: “Emergency services should implement specific protocols for the triage of mental health and substance-related calls, and develop multidisciplinary teams to help mitigate associated risks.”
Author Response

(The authors gave the same response as above.)

Round 2
Reviewer 2 Report
Comments and Suggestions for Authors
In my view, the revisions made to the article have significantly improved its clarity and scientific rigour, making it suitable for acceptance and publication in its current form.